# Safety and effectiveness of stereotactic radiosurgery for larger hemangioblastomas (>2cc): A multi-center retrospective study

Pavel S. Pichardo-Rojas[1☉], Salem M Tos[2☉], Georgios Mantziaris[2], Ahmed Shaaban[2], Ahmed M. Nabeel[3,4], Wael A. Reda[3,5], Sameh R. Tawadros[3,5], Khaled AbdelKarim[3,6], Amr M. N. El-Shehaby[3,5], Reem M Emad[3,7], Zhishuo Wei[8], Lindsay M. McKendrick[8], Ajay Niranjan[8], L. Dade Lunsford[8], Selcuk Peker[9], Yavuz Samanci[9], Roman Liscak[10], Jaromir May[10], David Mathieu[11], Cheng-chia Lee[12,13], Huai-che Yang[12,13], Antonio Dono[1], Angel I. Blanco[1], Yoshua Esquenazi[1], Nuria Martinez Moreno[14], Roberto Martinez Álvarez[14], Piero Picozzi[15], Andrea Franzini[15], Manjul Tripathi[16], Takuma Sumi[17], Takeo Uzuka[17], Hideyuki Kano[17], David Bailey[18], Brad E. Zacharia[18], Christopher P. Cifarelli[19,20], Daniel T. Cifarelli[19], Joshua D. Hack[20], Herwin Speckter[21], Erwin Lazo[21], Ronald E. Warnick[22], Jonathan E. Schoenhals[23], Joshua D. Palmer[23], Zhiyuan Xu[2], Jason P. Sheehan[2]*

1 Vivian L. Smith Department of Neurosurgery, University of Texas Health Science Center at Houston, Texas, United States of America, 2 Department of Neurological Surgery, University of Virginia, Charlottesville, Virginia, United States of America, 3 Gamma Knife Center Cairo, Nasser Institute Hospital, Cairo, Egypt, 4 Department of Neurological Surgery, Benha University, Qalubya, Egypt, 5 Department of Neurological Surgery, Ain Shams University, Cairo, Egypt, 6 Department of Clinical Oncology, Ain Shams University, Cairo, Egypt, 7 Department of Radiation Oncology, National Cancer Institute, Cairo University, Cairo, Egypt, 8 Department of Neurological Surgery, University of Pittsburgh medical center, Pittsburgh, Pennsylvania, United States of America, 9 Department of Neurological Surgery, Koc University School of Medicine, Istanbul, Turkey, 10 Department of Stereotactic and Radiation Neurosurgery, Na Homolce Hospital, Prague, Czech Republic, 11 Department of Neurological Surgery, Université de Sherbrooke, Centre de recherche du CHUS, Canada, 12 Department of Neurological Surgery, Neurological Institute, Taipei Veteran General Hospital, Taipei, Taiwan, 13 School of Medicine, National Yang Ming Chiao Tung University, Taipei, Taiwan, 14 Radiosurgery Unit, Hospital Ruber Internacional, Madrid, Spain, 15 Department of Neurological Surgery, IRCCS Humanitas Research Hospital, Milan, Italy, 16 Department of Neurological Surgery, Postgraduate Institute of Medical Education and Research, Chandigarh, India, 17 Department of Neurological Surgery, Dokkyo Medical University, Mibu, Tochigi, Japan, 18 Department of Neurological Surgery, Penn State Health - Hershey Medical Center, Pennsylvania, United States of America, 19 Department of Neurological Surgery, West Virginia University, United States of America, 20 Department of Radiation Oncology, West Virginia University, United States of America, 21 Dominican Gamma Knife Center and Radiology Department, CEDIMAT, Santo Domingo, Dominican Republic, 22 Gamma Knife Center, Jewish hospital, Mayfield clinic, Cincinnati, Ohio, United States of America, 23 Department of Radiation Oncology, The Ohio State University, Ohio, United States of America

☉ These authors contributed equally to this work.
* JPS2F@uvahealth.org

## Abstract

### Introduction

Hemangioblastomas (HGB) are the most common primary intra-axial tumors in the posterior fossa in adults, with an overall occurrence of 7–10%. They occur sporadically or as part of von Hippel-Lindau (VHL) disease. The role of stereotactic

or otherwise used by anyone for any lawful purpose. The work is made available under the Creative Commons CC0 public domain dedication.

**Data availability statement:** The data underlying the results presented in this study are not publicly available due to ethical and legal restrictions related to patient confidentiality. Researchers who meet criteria for access may contact Professor David Schlesinger, PhD (DJS9C@uvahealth.org) at the University of Virginia Medical Center. The International Radiosurgery Research Foundation (IRRF) maintains the dataset (https://www.irr-f.org/).

**Funding:** The author(s) received no specific funding for this work.

**Competing interests:** The authors have declared that no competing interests exist.

radiosurgery (SRS) as a minimally invasive treatment in larger HGB (>2cc) has not been thoroughly investigated.

## Methods

This multi-center study retrospectively analyzed data from 91 patients with large HGB (>2cc) treated between 1993 and 2023. Patients were stratified into VHL-associated and sporadic groups, with assessments including radiosurgical parameters, tumor response, overall survival (OS), and progression-free survival (PFS).

## Results

Patients with VHL-associated HGB were younger at diagnosis (median: 33 years vs. 52 years, p<0.001) and presented more frequently with multiple tumors (68.8% vs. 23.2%, p<0.001). Cerebellar lesions were the most common location (70%), followed by brainstem lesions (21%). The median target tumor volume was smaller in VHL cases (3.49 cc vs. 6.5 cc, p=0.038). Tumor control was achieved in 70% of cases across groups, with no significant differences in outcomes between VHL and sporadic cases. OS (170 months for VHL and 199 months for sporadic cases) and PFS (108 months for both groups) were comparable. Radiation necrosis was observed in 8.8% of patients.

## Conclusions

SRS may provide favorable tumor control with low morbidity in both VHL-associated and sporadic cases of larger HGB (>2 cc).Future studies should compare SRS with resection for larger HGB and explore molecular predictors of favorable response to SRS.

## Introduction

Hemangioblastomas (HGB), while uncommon, they represent the most common primary posterior fossa intra-axial tumor in adults [1], with an overall occurrence of 7−10% in this location [2,3]. These slow-growing lesions can occur in the cerebellum, brainstem, or spinal cord. HGB occur either sporadically or as part of von Hippel-Lindau (VHL) disease, an autosomal-dominant genetic syndrome characterized by the development of multiple tumors [4]. The pathogenesis of HGB involves inactivation of the *VHL* tumor suppressor gene, which is implicated in both sporadic and VHL-associated cases. In sporadic HGB, somatic mutations or allelic deletions of the *VHL* gene are found in up to 50% of cases [5–7]. Additionally, gain-of-function mutations in *hypoxia-inducible factor-2 alpha (HIF2A)* [8], regulated by the VHL gene, have been reported in sporadic tumors, further highlighting the molecular underpinnings of this disease.

Sporadic HGB generally present as solitary lesions, while VHL-associated cases are often multifocal and diagnosed at a younger age [9,10]. These tumors can significantly impact neurological function through direct compression of neural structures or secondary effects such as hydrocephalus [11,12]. Their potential for sudden, severe

complications, such as acute hemorrhage, further compounds their clinical impact and may necessitate emergency intervention [13].

While resection remains the standard of care for large, surgically-accessible HGB, stereotactic radiosurgery (SRS) has emerged as an effective alternative for deep-seated lesions, those located near eloquent regions, or cases with high surgical risk [12,14]. By delivering precise, high-dose radiation, SRS aims to achieve tumor control while minimizing damage to surrounding critical brain structures. Despite its established role in smaller lesions, the use of SRS for managing large HGB exceeding 2cc in volume has not been thoroughly investigated [10]. To address this gap, we conducted the first multi-institutional study evaluating the safety and efficacy of SRS in larger HGB.

## Methods

### Patient population and inclusion criteria

Seventeen centers from the International Radiosurgery Research Foundation (IRRF) participated in this retrospective, multicenter study, providing data on patients treated between 1993 and 2023. The initial database included 193 patients and 570 SRS-treated tumors. Tumor volume measurements were performed using post-contrast T1-weighted MRI sequences, including both the solid enhancing nodule and associated cyst, when present. In cases where exact volumetric data were unavailable, tumor volume was estimated based on measured tumor dimensions. To focus on larger lesions, only tumors exceeding 2 cc were included, a threshold determined by the principal investigators at each center, reflecting clinical relevance and supported by prior literature [15]. For patients with multiple tumors, only the largest tumor was retained in the analysis. Following these criteria, data from 14 centers were included, comprising a total of 91 patients with 91 large HGB exceeding 2cc. Each center was responsible for its own data collection and securing IRB approval. Patient consent was not required due to the retrospective nature of the study. Radiological and clinical data were shared through a de-identified database.

This retrospective, multicenter study was conducted in collaboration with 14 centers participating in the International Radiosurgery Research Foundation (IRRF). Each participating institution obtained approval from its respective Institutional Review Board (IRB) or ethics committee prior to data collection. Due to the retrospective nature of the study and the use of de-identified data, patient consent was not required. All data were anonymized at the site level before submission to the central database. This study complies with the ethical standards outlined in the Declaration of Helsinki and adheres to the STROBE (Strengthening the Reporting of Observational Studies in Epidemiology) guidelines [16]. Data collection was conducted by co-authors across multiple centers between September 1, 2023, and January 20, 2024. The IRRF coordinator and the first author had access to identifiable participant information during and after the data collection period.

SRS was delivered either as a primary treatment or as adjuvant/salvage therapy for residual tumors. All included patients had at least one clinical and radiologic follow-up, typically performed within six months after SRS.

### Radiosurgical technique

Each center utilized the radiosurgery technology available at their facility at the time of the procedure. This included Gamma Knife units appropriate for the treatment period and center. Typically, a stereotactic frame was placed, followed by acquisition of high-resolution, stereotactic magnetic resonance imaging (MRI) or computed tomography (CT) scans. The radiosurgical plan was then developed and approved by the site's multidisciplinary team. All SRS procedures in this study were performed in a single session [9].

### Clinical and radiological follow up

Patients were typically followed clinically and radiographically with gadolinium-enhanced MRI every 6 months after the initial SRS and then at 6-to-12-month intervals, with further follow-up adjusted longitudinally based on individual patient needs and the protocols of each participating center.

 

Overall survival (OS) was defined as the time from the date of initial clinical diagnosis to death from any cause. Progression-free survival (PFS) was defined as the time from SRS following diagnosis, to radiographic progression, as outlined above.

Tumor response was categorized as regression (≥20% decrease in tumor volume from baseline), stability (±20% change in tumor volume from baseline), or progression (≥20% increase in tumor, cyst volume, or both from baseline) [9], per standardized consensus across centers and utilizing previously implemented protocols [17]. Tumor control was defined as either regression or stability.

Adverse radiation effects (AREs) were defined as new radiographic or clinical changes attributable to SRS, graded according to the Radiation Therapy Oncology Group (RTOG) toxicity criteria [18].

### Statistical analysis

Patients were divided into two groups: VHL and sporadic HGB. Categorical variables were compared using Pearson's chi-square tests, while non-parametric data were analyzed with Mann-Whitney U and Kruskal-Wallis tests. Survival outcomes were evaluated through log-rank tests for univariate analysis and Cox regression for both univariate and multivariate analysis, incorporating covariates that showed statistical significance in the univariate analysis. A p-value of <0.05 was considered statistically significant, with a corresponding 95% confidence interval. All analyses were performed using IBM SPSS Statistics version 25.0 (IBM Corp., Armonk, NY).

## Results

### Patient characteristics

This multi-center study included 91 patients with larger HGB lesions exceeding 2cc, comprising 35 VHL-associated cases and 56 sporadic cases. Patients with VHL were significantly younger at diagnosis (median age 33 years) compared to sporadic cases (median age 52 years, p < 0.001) and at the time of SRS (median age 42 vs. 55 years, p < 0.001). Large HGB were more frequently found in newly diagnosed VHL cases (45.7% vs. 16.1%, p = 0.002). Conversely, larger HGB were more commonly observed in recurrent sporadic cases (35.7% vs. 14.3%, p = 0.026). Similarly, larger HGB were more frequently associated with multiple tumors in VHL cases (68.8%) compared to sporadic cases (23.2%, p < 0.001). Tumor location varied slightly between VHL-associated and sporadic cases, but no statistically significant differences were observed (p = 0.282). The majority of tumors in both groups were located in the cerebellum, comprising 75.8% of VHL cases and 69.6% of sporadic cases. Brainstem tumors were more common in sporadic cases (15.7%) compared to VHL-associated cases (5.6%). Parietal tumors were only observed in the VHL group (6.1%), while occipital tumors were exclusive to the sporadic group (1.8%). Temporal tumors occurred at a similar rate in both groups (3% in VHL vs. 3.6% in sporadic cases). Imaging characteristics including solid, cystic, or mixed tumors, were similar between groups (p = 0.928). The median follow-up duration was comparable (51.5 vs. 58 months, p = 0.893). Baseline demographics characteristics are detailed in **Table 1**.

### Radiosurgical parameters

The median target tumor volume was smaller in VHL-associated tumors (3.49 cc vs. 6.5 cc, p = 0.038), while the number of isocenters was higher in sporadic cases (17 vs. 12, p < 0.001). Both groups received similar median margin doses (15 Gy vs. 14.5 Gy, p = 0.825) and maximum doses (28 Gy vs. 28 Gy, p = 0.923). Radiosurgical parameters are detailed in **Table 2**.

### Tumor response after SRS

Tumor response rates were comparable between groups. Tumor regression was observed in 39.4% of VHL cases and 45.5% of sporadic cases, while tumor stability was achieved in 39.4% of VHL cases and 20% of sporadic cases

**Table 1. Baseline patient characteristics.**

| Characteristics | VHL, N = 35 | Sporadic, N = 56 | p-value |
|---|---|---|---|
| **Gender (Male)** | 21 (60%) | 33 (58.9%) | 0.919 |
| **Median age at SRS (IQR)** | 42 (31, 44) | 55 (45, 64) | **<0.001** |
| **Median age at diagnosis (IQR)** | 33 (22, 43) | 52 (41, 59.7) | **<0.001** |
| **Median duration of Symptoms before SRS in months (IQR)** | 9 (4, 33) | 6 (3, 26) | 0.413 |
| **Median duration from SRS to last follow-up in months (IQR)** | 51.5 (27, 93) | 58 (29, 97) | 0.893 |
| **Indication of SRS** | | | |
| newly developed | 16 (45.7%) | 9 (16.1%) | **0.002** |
| recurrent | 5 (14.3%) | 20 (35.7%) | **0.026** |
| residual | 13 (37.1%) | 30 (53.6%) | 0.127 |
| **Tumor Location** | | | 0.282 |
| Cerebellum | 25 (75.8%) | 39 (69.6%) | |
| Parietal | 2 (6.1%) | 0 (0%) | |
| Temporal | 1 (3%) | 2 (3.6%) | |
| Occipital | 0 (0%) | 1 (1.8%) | |
| Brainstem | 5 (5.6%) | 14 (15.7%) | |
| **Status of tumor number** | | | **<0.001** |
| multiple | 24 (68.6%) | 13 (23.2%) | |
| single | 11 (31.4%) | 43 (76.8%) | |
| **Tumor imaging characteristics** | | | 0.928 |
| Solid | 13 (41.9%) | 17 (42.5%) | |
| Cystic | 12 (38.7%) | 14 (35%) | |
| Mixed | 6 (19.4%) | 9 (22.5%) | |
| **Median KPS at SRS (IQR)** | 90 (80, 90) | 90 (80, 90) | 0.819 |
| **Number of prior cranial resections (IQR)** | 2 (1, 2) | 1 (1, 2) | 0.053 |
| **Treatment prior to SRS** | | | 0.321 |
| no treatment | 1 (0%) | 0 (0%) | |
| Fractionated radiation therapy | 2 (7.4%) | 1 (2%) | |
| resection | 19 (70.4%) | 38 (74.5%) | |
| Fractionated radiotherapy & resection | 5 (18.5%) | 12 (15.4%) | |
| **Further treatments following failed SRS** | | | 0.814 |
| SRS | 4 (11.4%) | 10 (17.8%) | |
| Resection | 3 (8.5%) | 7 (12.5%) | |
| Fractionated radiation therapy | 0 (0%) | 1 (1.78%) | |

(p = 0.119). Tumor control (defined as either regression or stability) was achieved in 78.8% of VHL cases and 65.5% of sporadic cases. The log-rank test showed no significant difference between groups (p = 0.28). The Kaplan-Meier curve illustrating tumor control is presented in **Fig 1**. Three out of 91 patients have not had a follow up MRI to assess SRS response.

Local tumor control in larger HGB was not influenced by etiology, regardless of whether the tumors were VHL-associated or sporadic (OR 1.96, 95% CI: 0.71–5.34, p = 0.188), nor by gender (OR 1.01, 95% CI: 0.39–2.58, p = 0.983), age at diagnosis (OR 1.01, 95% CI: 0.98–1.04, p = 0.467), age at SRS (OR 0.917, 95% CI: 0.751–1.122, p = 0.400), tumor volume (OR 1.014, 95% CI: 0.926–1.109, p = 0.770), KPS at SRS (OR 1.034, 95% CI: 0.969–1.103, p = 0.309), symptom duration (OR 1.014, 95% CI: 0.992–1.036, p = 0.217), or presence of multiple tumors (OR 0.404, 95% CI: 0.054–3.002, p = 0.376).

**Table 2. SRS treatment characteristics and clinical outcomes.**

| Characteristics | VHL, N = 35[1] | Sporadic, N = 56[1] | p-value |
|---|---|---|---|
| **Target Tumor volume (cc)** | 3.49 (2.5, 7.5) | 6.5 (3.55, 11.48) | **0.038** |
| **Number of Fractions** | 1 (1, 1) | 1 (1, 1) | 0.275 |
| **Margin Dose (Gy)** | 15 (13, 17) | 14.5 (13, 16) | 0.825 |
| **Maximum Dose (Gy)** | 28 (26, 34) | 28 (25, 33.6) | 0.923 |
| **Number of isocenters** | 12 (7, 13) | 17 (9, 26) | **<0.001** |
| **Tumor response** | | | 0.119 |
| Stable | 13 (39.4%) | 11 (20%) | |
| Regression | 13 (39.4%) | 25 (45.5%) | |
| Progression | 7 (21.2%) | 19 (34.5%) | |
| Cystic Enlargement | 2 (5.7%) | 8 (14.2%) | 0.080 |
| **Adverse radiation effects** | 3 (8.5%) | 5 (8.9%) | 0.944 |

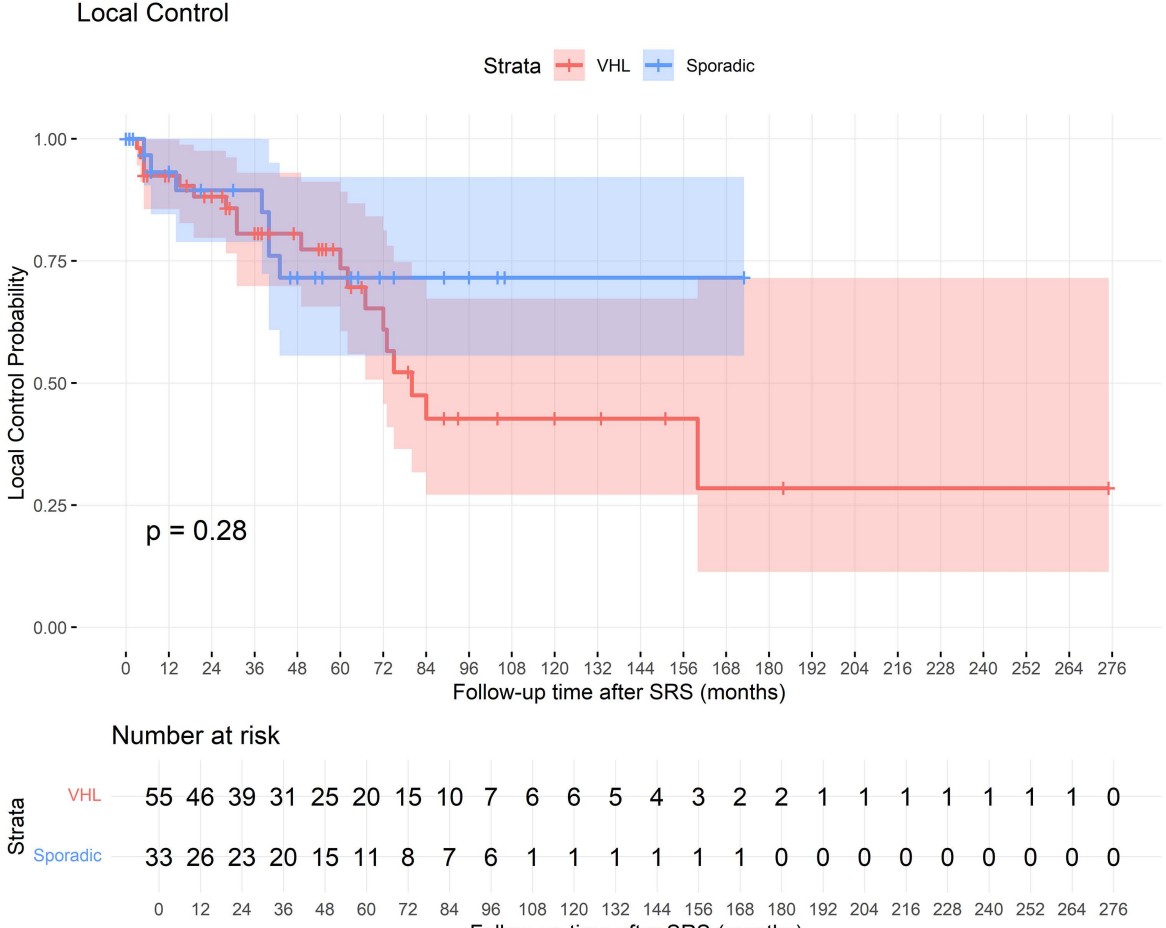

**Fig 1. Kaplan-Meier Curve for Local Tumor Control.**

## Post SRS Complications

Adverse radiation effects (ARE's) were infrequent and experienced in a total of eight patients (8.7%), the distribution was similar between groups, occurring in 8.5% of VHL cases and 8.9% of sporadic cases (p=0.944). Classified retrospectively using the RTOG toxicity criteria [18]: 6 as Grade 1 (asymptomatic) and 2 as Grade 2 (symptomatic, managed conservatively); among those two, one reported imbalance, while the other experienced headache, nausea, and dizziness. Cystic enlargement was more common in sporadic cases (14.2% vs. 5.7%), but this difference was not statistically significant (p=0.080).

Patients with ARE's had a higher margin dose (median 16 Gy [IQR: 16, 19.5] vs. 14.5 Gy [IQR: 13, 16.7], p=0.026) and maximum dose (median 34 Gy [IQR: 29.9, 40] vs. 27.6 Gy [IQR: 24.5, 32.9], p=0.047). In contrast, tumor volume, number of SRS fractions, number of isocenters, age at diagnosis, age at SRS, duration of symptoms, presence of multiple tumors, recurrent tumors, etiology, or gender did not show any correlation with ARE's (p>0.05).

## Survival Outcomes

The mean OS was 170 months for VHL-associated cases and 199 months for sporadic cases, with no significant difference between groups (log-rank p=0.11). The Kaplan-Meier curve illustrating OS is presented in **Fig 2**. Notably, among the 11 deceased patients in our cohort, only one death was directly attributed to HGB, while the remaining 10 deaths were unrelated to HGB.

The OS hazard ratio (HR) in sporadic cases relative to VHL was 0.62 (95% CI: 0.19–1.93, p=0.412), Older age at diagnosis (HR 1.05 [95% CI: 1.00–1.09], p=0.016) and at SRS (HR 1.04 [95% CI: 1.00–1.08], p=0.031) were associated with worse OS, although neither factor retained statistical significance in multivariate analysis. Other factors, including gender (p=0.776), KPS at SRS (p=0.165), cyst presence at diagnosis (p=0.387), the presence of multiple tumors (p=0.295), tumor volume (p=0.611), and duration of symptoms (p=0.997), did not significantly influence OS.

The mean PFS was 108 months in both VHL-associated and sporadic cases, with no significant difference between groups (log-rank p=0.35). The Kaplan-Meier curve illustrating PFS is presented in **Fig 3**. Similar to OS, older age at diagnosis (HR 1.02 [95% CI: 1.00–1.04], p=0.041) and at the time of SRS (HR 1.02 [95% CI: 1.00–1.04], p=0.018) were associated with worse PFS; however, these associations did not persist in multivariate analysis. Sporadic cases, compared to VHL-associated lesions, did not significantly impact PFS (HR 1.66 [95% CI: 0.84–3.31], p=0.144). Additionally, gender (p=0.498), KPS at SRS (p=0.381), cyst presence at diagnosis (p=0.377), presence of multiple tumors (p=0.274), tumor volume (p=0.673), and symptom duration (p=0.380) were not significantly associated with PFS. The HR analyses are depicted in **Table 3**.

## Salvage treatment after initial SRS for larger HGB

The majority of our cohort (98.9%) received treatment prior to SRS. Specifically, one patient received no prior treatment, three had prior radiation therapy, 57 underwent prior resection, and 17 with both radiotherapy and resection. These distributions were similar between VHL and sporadic groups (p=0.321). Following the index SRS for study accrual to a larger HGB, 15% required additional SRS, 10.9% underwent resection, and 0.9% required fractionated radiotherapy, with these outcomes being relatively evenly distributed across the cohort (p=0.814).

## Discussion

The present study is the most comprehensive evaluation of larger (>2cc) HGB in a multicenter cohort of 91 patients treated across 14 centers. Patients with VHL-associated HGB were younger at both diagnosis and SRS compared to those with sporadic HGB. Larger HGB were more commonly observed in newly developed tumors of VHL cases, whereas larger recurrent HGB were predominantly found in sporadic cases. The cerebellum (70%) and brainstem (21%) were the most frequent tumor locations in both groups, with multiple tumors being significantly more common in VHL cases. Sporadic tumors required larger target volumes and a greater number of isocenters during SRS planning. A favorable tumor

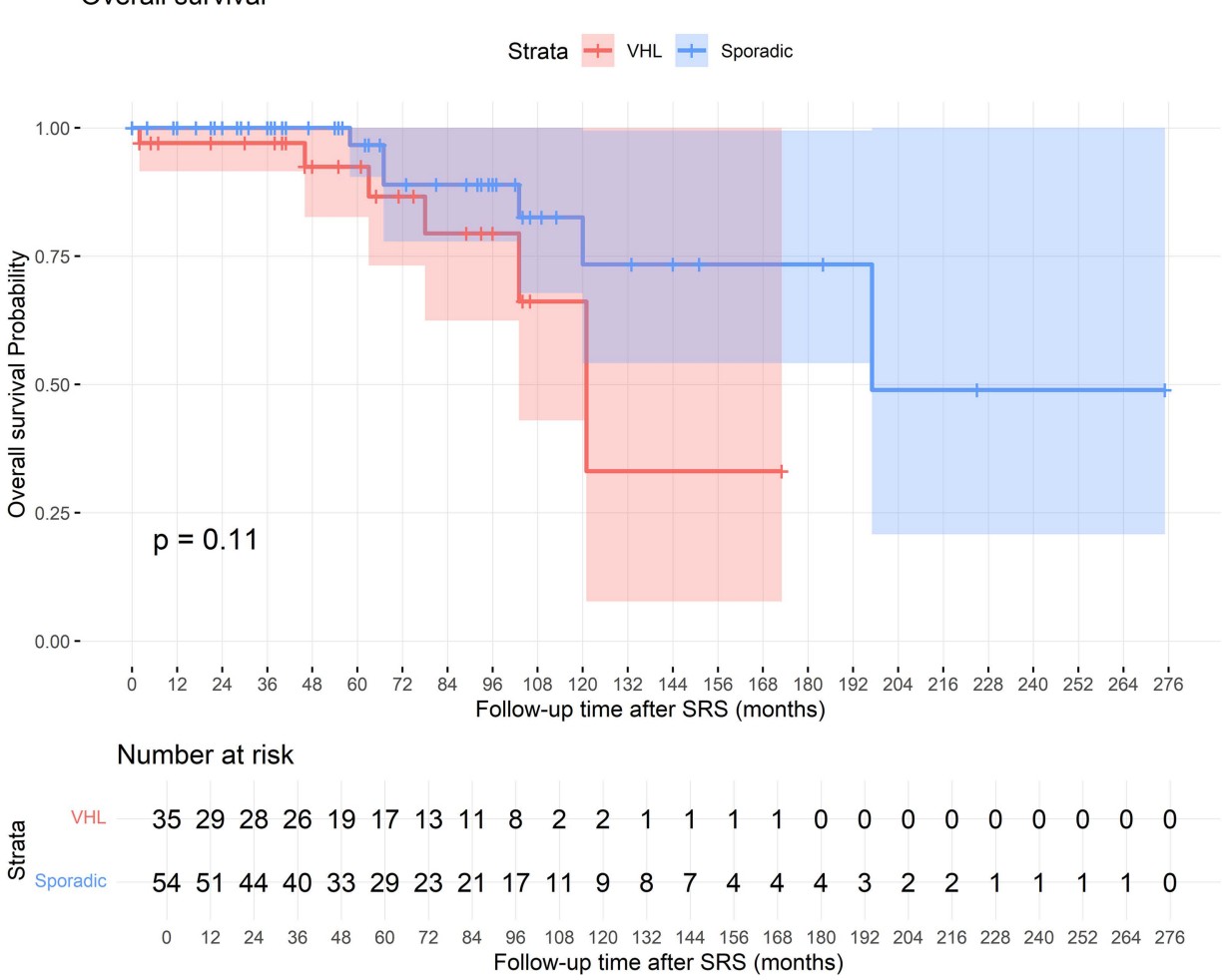

Fig 2. Kaplan-Meier Curve for Overall Survival.

response, defined as regression or stable disease, was observed in approximately 70% of cases at last follow up, with similar distribution between VHL-associated and sporadic groups. ARE's occurred in 8.7% of patients, with two symptomatic cases, and similar rates were observed between VHL and sporadic groups. OS and PFS were comparable between the groups, with a mean OS of 170 months for VHL cases and 199 months for sporadic cases. Similarly, the mean PFS was 108 months in both groups, reflecting consistent long-term outcomes regardless of tumor etiology. These results underscore the safety and effectiveness of SRS in managing large HGB in both VHL-associated and sporadic cases.

These results align with previous reports underscoring the clinical utility of SRS for smaller lesions but expand its applicability to larger tumors, a subset that has been poorly studied in the literature [9]. Similar to patients with smaller lesions, those with VHL-associated HGB in this study were diagnosed at a significantly younger age and more frequently presented with multiple tumors compared to their sporadic counterparts. This finding raises the possibility that VHL-associated HGB may not only occur earlier but could also present with larger lesions more commonly than sporadic cases. This pattern likely reflects the well-established genetic predisposition and clinical phenotype of VHL disease [4,19,20], which often includes the development of multiple synchronous or metachronous tumor growths over time.

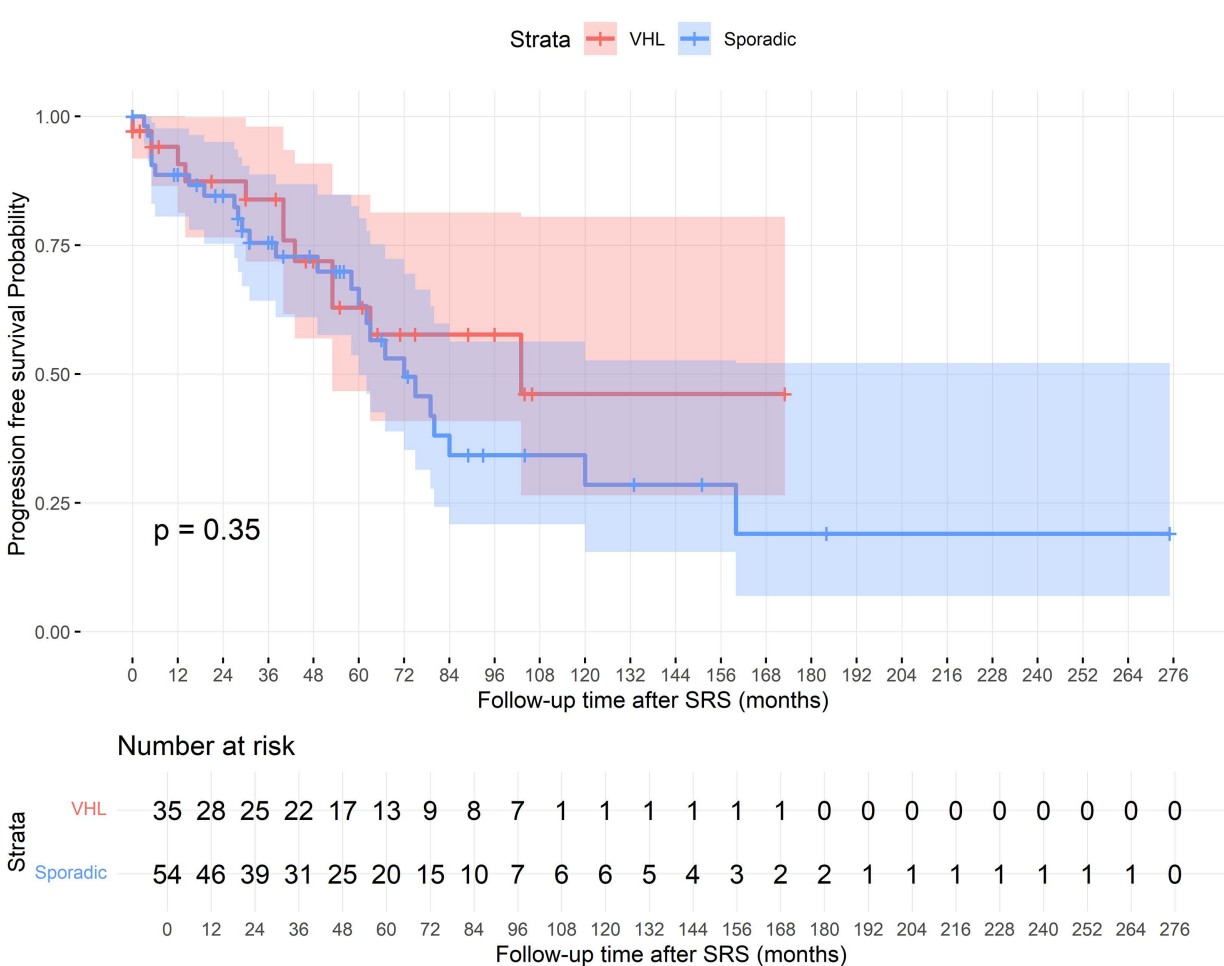

Progression Free Survival

**Fig 3. Kaplan-Meier Curve for Progression-Free Survival.**

   In younger patients with a family history of VHL, physicians should remain vigilant for symptoms that could be associated with mass effect from larger HGB, such as headache, cerebellar dysfunction, or spinal cord compression [21–23]. These symptoms warrant prompt diagnostic workup, as our findings indicate that even VHL-associated larger HGB can be managed safely and effectively with SRS, achieving favorable outcomes while maintaining low morbidity and mortality. In contrast, larger sporadic HGB were more often solitary and more frequently occurred as recurrences later in life.

   While small HGB (<10 mm) may appear isointense on T1-weighted images and hyperintense on T2-weighted images with homogeneous contrast enhancement [24], larger HGB typically present as an intra-axial cystic mass with an enhancing mural nodule abutting the pia or as a solid, intensely enhancing mass, often accompanied by flow-voids from dilated vessels adjacent to or within the tumor. This distinction underscores the importance of individualized diagnostic and management strategies tailored to the unique characteristics and clinical contexts of these tumors.

   In our cohort, tumor location varied, with cerebellar lesions accounting for 70% of total cases, making them the most common location regardless of etiology. This finding aligns with the literature, which reports an incidence of cerebellar HGB ranging from 50% to 80% [21,22,25–27]. Conversely, larger brainstem HGB were observed in 21% of cases in this

**Table 3. Univariable and multivariable Cox-regression.**

| Characteristic | Overall Survival | | | | | |
| | Univariable | | | Multivariable | | |
| | HR | 95% CI | p-value | HR | 95% CI | p-value |
|---|---|---|---|---|---|---|
| Sporadic (Reference = VHL) | 0.62 | 0.19, 1.93 | 0.412 | – | – | – |
| Male Gender | 1.18 | 0.37, 3.79 | 0.776 | – | – | – |
| Age at Diagnosis | 1.05 | 1.00, 1.09 | **0.016** | 1.05 | 0.95, 1.15 | 0.304 |
| Age at SRS | 1.04 | 1.00, 1.08 | **0.031** | 0.99 | 0.91, 1.09 | 0.971 |
| KPS at SRS | 0.97 | 0.95, 1.00 | 0.165 | – | – | – |
| Cyst present at Diagnosis | 0.53 | 0.13, 2.18 | 0.387 | – | – | – |
| Multiple tumors | 0.44 | 0.09, 2.04 | 0.295 | – | – | – |
| Tumor Volume (cc) | 0.97 | 0.86, 1.08 | 0.611 | – | – | – |
| Duration of Symptoms | 1.00 | 0.97, 1.02 | 0.997 | – | – | – |
| Characteristic | Progression-Free Survival | | | | | |
| | Univariable | | | Multivariable | | |
| | HR | 95% CI | p-value | HR[1] | 95% CI | p-value |
| Sporadic (Reference = VHL) | 1.66 | 0.84, 3.31 | 0.144 | – | – | – |
| Male Gender | 1.25 | 0.64, 2.44 | 0.498 | – | – | – |
| Age at Diagnosis | 1.02 | 1.00, 1.04 | **0.041** | 0.98 | 0.94, 1.03 | 0.633 |
| Age at SRS | 1.02 | 1.00, 1.04 | **0.018** | 1.03 | 0.98, 1.09 | 0.179 |
| KPS at SRS | 1.00 | 0.98, 1.03 | 0.381 | – | – | – |
| Cyst present at Diagnosis | 0.70 | 0.31, 1.56 | 0.377 | – | – | – |
| Multiple tumors | 0.67 | 0.33, 1.36 | **0.041** | 0.76 | 0.36, 1.59 | 0.473 |
| Tumor Volume (cc) | 1.01 | 0.96, 1.06 | 0.673 | – | – | – |
| Duration of Symptoms | 1.00 | 0.99, 1.01 | 0.380 | – | – | – |

HR = Hazard Ratio, CI = Confidence Interval

**\*** SRS = Stereotactic radiosurgery, KPS = Karnofsky performance score, VHL = Von Hippel-Lindau

cohort, which apparently occur more frequently in larger tumors compared to their smaller counterparts, which have been reported to occur in 7% to 10% of cases [9,25,26]. However, no direct comparisons with smaller lesions were performed. Larger brainstem lesions pose significant challenges due to their proximity to critical structures. While meta-analyses of resection for brainstem HGB report gross total resection rates as high as 98% with a 4% mortality rate, these studies have not specifically assessed large HGB [28]. In this context, our study suggest that SRS could be utilized safely, and could potentially represent a potential treatment option to resection, especially given the risks of open resection for large vascular lesions; however, this study did not make parallel assessments with surgical cohorts [28]. Nevertheless, direct comparisons between cerebellar and brainstem lesions treated with SRS versus resection [29–31], including adjunctive measures such as embolization [32,33], are essential to refine management strategies. Although SRS complications are likely limited, future studies should evaluate these outcomes to guide treatment decisions more comprehensively.

Tumor control at last follow up was achieved in over 70% of cases across groups, defined as either regression or stability, with homogenous outcomes between VHL-associated and sporadic cases. These results align with prior studies of SRS for HGB [9,34,35], which report control rates exceeding 80% at five years for smaller lesions. Our findings primarily

reflect the use of single-fraction SRS for tumor volumes averaging 3.5 cc in VHL-associated cases and 6.5 cc in sporadic cases, with a mean margin dose of 15 Gy and a maximum dose of 28 Gy. In our cohort, a higher margin dose (median 16 Gy vs. 14.5 Gy) and maximum dose (median 34 Gy vs. 27.6 Gy) were associated with the presence of ARE's. While these parameters yielded a radiation necrosis rate of 8.8%, irrespective of etiology, this is consistent with rates reported for smaller lesions, estimated at approximately 7% [9,35,36]. Hypofractionated SRS may be considered in cases involving larger tumor volumes to further mitigate risks. Notably, only 2 out of 8 patients with AREs were symptomatic, highlighting the overall safety of this intervention. These findings underscore that even larger lesions, previously deemed less well suited for SRS, can be effectively managed with this SRS.

Survival outcomes, measured as OS and PFS, were comparable between VHL-associated and sporadic cases. The mean OS of 170 months in VHL patients and 199 months in sporadic cases underscores the indolent nature of these tumors [26,37–39], even in the context of larger volumes, and reflects a behavior similar to their smaller lesion counterparts. Notably, among the 11 deceased patients in our cohort, only one death was directly attributed to HGB, while the remaining 10 deaths were unrelated to HGB, further emphasizing the typically slow progression of these tumors. While cause-specific survival was not influenced by VHL or sporadic etiology in large HGB, age at diagnosis and at the time of SRS were predictors of worse OS and PFS. These findings highlight the critical importance of early diagnosis to optimize patient outcomes. Notably, gender, etiology, tumor volume beyond 2cc, presence of cysts at diagnosis, and duration of symptoms did not significantly impact survival outcomes in this cohort.

While supratentorial HGB have been reported to be safely treated with resection, incomplete resection is more commonly observed in some locations such as the sellar/suprasellar region [40]. In surgically inaccessible or eloquent areas where a gross total resection is deemed unlikely, SRS could be considered a viable treatment option. However, future studies are needed to further evaluate its efficacy and safety in managing large HGB. This is particularly significant for VHL patients, who often face repeated interventions due to multiple lesions [4,41]. Nevertheless, while our findings suggest that SRS may be implemented for larger HGB, the tumor control rate in both groups still leaves a substantial proportion of patients requiring salvage treatment, highlighting the potential role of multimodal approach.

In summary, while these results are encouraging, further studies, including randomized trials, are necessary to validate our findings and refine treatment strategies for large HGB.

## Limitations

While this multicenter study extensively addresses a previously unexplored question using strict inclusion criteria across a large cohort, it is not without limitations inherent to its retrospective design. Despite efforts to standardize data collection, screening, and selection processes, potential selection bias, variability in data collection across centers, and differences in radiosurgical techniques remains. Although strict inclusion criteria were applied, the heterogeneity of SRS equipment and protocols, as well as the learning curve across the long study period (1993–2023), may have influenced treatment outcomes. While we achieved a robust median follow-up duration of 51.5 to 58 months, three cases were lost to follow-up. Furthermore, although the study stratifies outcomes between VHL-associated and sporadic HGB, it does not include molecular analyses, such as *VHL* or *HIF2A* mutation status, which could provide valuable insights into treatment response. Additional limitations include the absence of central imaging review, the potential for misclassification of progression versus pseudoprogression, lack of quality-of-life or functional outcome measures, and heterogeneity in prior treatments and timing of SRS. While we implemented strict criteria for tumor assessment, isolated cyst expansion may reflect vascular permeability–driven changes rather than true progression, and the absence of stratified volumetric analysis between cystic and solid components over time may limit interpretability. Lastly, the absence of direct comparisons with surgical outcomes limits our ability to draw conclusions on the relative efficacy of SRS, particularly for surgically accessible lesions. Overall, this study provides a foundation for future prospective, multi-institutional clinical trials.

## Conclusion

In this multi-institutional study, SRS may offer favorable tumor control rates with low morbidity in both VHL-associated and sporadic cases of larger HGB(>2cc). Comparable OS and PFS outcomes between groups highlight its viability as a minimally invasive alternative, particularly for surgically challenging lesions. Future research should explore direct comparisons with resection and incorporate molecular analyses to refine treatment strategies further.

## Author contributions

**Conceptualization:** Pavel S. Pichardo-Rojas, Jason P. Sheehan.

**Data curation:** Pavel S. Pichardo-Rojas, SALEM M TOS, Ahmed Shaaban, Ahmed M. Nabeel, Wael A. Reda, Sameh R. Tawadros, Khaled AbdelKarim, Amr M. N. El-Shehaby, Reem M Emad, Zhishuo Wei, Lindsay M. McKendrick, Ajay Niranjan, L Dade Lunsford, Selcuk Peker, Yavuz Samanci, Roman Liscak, Jaromir May, David Mathieu, Cheng-chia Lee, Huai-che Yang, Antonio Dono, Angel I. Blanco, Yoshua Esquenazi, Nuria Martinez Moreno, Roberto Martinez Álvarez, Piero Picozzi, Andrea Franzini, Manjul Tripathi, Takuma Sumi, Takeo Uzuka, Hideyuki Kano, David Bailey, Brad E. Zacharia, Christopher P. Cifarelli, Daniel T. Cifarelli, Joshua D. Hack, Herwin Speckter, Erwin Lazo, Ronald E. Warnick, Jonathan E Schoenhals, Joshua D Palmer, Zhiyuan Xu.

**Formal analysis:** Pavel S. Pichardo-Rojas, SALEM M TOS.

**Funding acquisition:** Pavel S. Pichardo-Rojas, SALEM M TOS.

**Investigation:** Pavel S. Pichardo-Rojas.

**Methodology:** Pavel S. Pichardo-Rojas, SALEM M TOS.

**Project administration:** Pavel S. Pichardo-Rojas.

**Resources:** Pavel S. Pichardo-Rojas.

**Software:** Pavel S. Pichardo-Rojas, SALEM M TOS.

**Supervision:** Jason P. Sheehan.

**Validation:** Pavel S. Pichardo-Rojas.

**Visualization:** Pavel S. Pichardo-Rojas, SALEM M TOS.

**Writing – original draft:** Pavel S. Pichardo-Rojas, Jason P. Sheehan.

**Writing – review & editing:** SALEM M TOS, Georgios Mantziaris, Ahmed Shaaban, Ahmed M. Nabeel, Wael A. Reda, Sameh R. Tawadros, Khaled AbdelKarim, Amr M. N. El-Shehaby, Reem M Emad, Zhishuo Wei, Lindsay M. McKendrick, Ajay Niranjan, L Dade Lunsford, Selcuk Peker, Yavuz Samanci, Roman Liscak, Jaromir May, David Mathieu, Cheng-chia Lee, Huai-che Yang, Antonio Dono, Angel I. Blanco, Yoshua Esquenazi, Nuria Martinez Moreno, Roberto Martinez Álvarez, Piero Picozzi, Andrea Franzini, Manjul Tripathi, Takuma Sumi, Takeo Uzuka, Hideyuki Kano, David Bailey, Brad E. Zacharia, Christopher P. Cifarelli, Daniel T. Cifarelli, Joshua D. Hack, Herwin Speckter, Erwin Lazo, Ronald E. Warnick, Jonathan E Schoenhals, Joshua D Palmer, Zhiyuan Xu, Jason P. Sheehan.

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
