## [Decision Letter · Decision Letter 0]

28 Aug 2025

Dear Dr. Sheehan,

Thank you for submitting your manuscript to PLOS ONE. After careful consideration, we feel that it has merit but does not fully meet PLOS ONE’s publication criteria as it currently stands. Therefore, we invite you to submit a revised version of the manuscript that addresses the points raised during the review process.

We look forward to receiving your revised manuscript.

Kind regards,

Wang-Rui Liu

Academic Editor

PLOS ONE

Journal Requirements:

Reviewers' comments:

Reviewer's Responses to Questions

**Comments to the Author**

1. Is the manuscript technically sound, and do the data support the conclusions?

Reviewer #1: Yes

Reviewer #2: Yes

2. Has the statistical analysis been performed appropriately and rigorously?

Reviewer #1: Yes

Reviewer #2: Yes

3. Have the authors made all data underlying the findings in their manuscript fully available?

Reviewer #1: Yes

Reviewer #2: Yes

4. Is the manuscript presented in an intelligible fashion and written in standard English?

Reviewer #1: Yes

Reviewer #2: Yes

Reviewer #1: This manuscript presents the largest multicenter series to date evaluating stereotactic radiosurgery (SRS) for larger hemangioblastomas (>2 cc), addressing an important gap in the radiosurgical literature. The study is technically sound in its core design, with clear inclusion criteria, a well-defined patient cohort, and appropriate primary outcomes of tumor control, overall survival (OS), and progression-free survival (PFS). The statistical methods employed, including Kaplan–Meier survival analysis, log-rank testing, and Cox regression, are appropriate for the data. Results are presented in an organized manner, and the overall writing is clear and intelligible.

However, several issues limit the strength of the conclusions. The retrospective nature of the study across fourteen centers inevitably introduces heterogeneity in SRS technique, imaging follow-up, and patient selection. While the authors acknowledge variability in equipment and protocols, the absence of any adjustment for inter-center or temporal differences in dose selection, imaging frequency, or margin dose may allow confounding effects on tumor control and adverse radiation event (ARE) rates. Given that the study period spans three decades (1993–2023), practice evolution during this interval could have influenced outcomes; stratifying results by treatment era or controlling for center in multivariable analysis would strengthen the validity of the findings.

The definition of tumor control as either regression or stability is reasonable, but the handling of cystic enlargement is not fully clarified. Hemangioblastomas often exhibit cyst dynamics independent of solid nodule growth, and it is important to state explicitly whether isolated cyst expansion was classified as progression, particularly since this may disproportionately affect larger lesions. The omission of volumetric progression-free survival analysis in such cases may limit interpretability. Additionally, while adverse radiation events were correlated with higher dose parameters, the finding that tumor control rates in this large-lesion cohort (~70%) are lower than historical reports for smaller lesions warrants more explicit discussion in both the abstract and conclusion, along with tempering the assertion that SRS is broadly “safe and effective” for all large lesions.

The absence of a surgical comparison arm limits the ability to contextualize SRS results relative to the current gold standard for accessible lesions. While the discussion refers to SRS as a potential alternative to resection, the lack of comparative data makes this claim premature. Statements implying equivalence should be revised to emphasize that SRS appears to be a viable option for surgically challenging cases, with low morbidity and acceptable control rates, but further direct comparisons are required. Similarly, although OS and PFS did not differ significantly between VHL-associated and sporadic cases, the interpretation should note that the tumor control rate in both groups still leaves a substantial proportion of patients requiring salvage treatment.

The manuscript is generally well written, but the abstract slightly overstates generalizability and should explicitly mention its retrospective design and lack of surgical comparator. Some redundancy between the Results and Discussion could be reduced, and minor grammatical edits would further improve flow. Figures and tables are appropriate, but table formatting could be standardized for decimal precision and unit presentation.

Finally, the data availability statement indicates that access requires institutional review board approval and a request to the corresponding author. While this is understandable given patient confidentiality, it is less than optimal in meeting PLOS ONE’s open data policy, and the feasibility of depositing anonymized data in a controlled-access repository should be explored.

In summary, this study provides valuable multicenter evidence supporting the role of SRS in managing larger hemangioblastomas, especially when surgery is not feasible. The conclusions are generally supported by the data but require a more cautious interpretation in light of the retrospective design, inter-center variability, and modest tumor control rates compared with smaller lesion series. With revisions to address confounding factors, clarify cystic lesion outcome classification, and adjust the framing of conclusions, this work would make a meaningful contribution to the literature.

Reviewer #2: I was invited to review a paper titled “Safety and effectiveness of stereotactic radiosurgery for larger hemangioblastomas (>2cc): a multi-center retrospective study” for PLOS ONE. This is a large multi-center retrospective study evaluating SRS for larger hemangioblastomas (>2cc), comparing VHL-associated and sporadic cases. The topic is clinically relevant and addresses a gap in the literature, as most prior SRS studies have focused on smaller lesions. The dataset is sizeable, spanning multiple international centers, and the results are clearly presented.

Please find my comments below:

The authors position this as the first multi-institutional study of large HGB (>2cc) treated with SRS, which is a relevant and novel scope. However, the introduction could better justify the chosen >2cc threshold, as this cut-off seems somewhat arbitrary without robust prior evidence.

The retrospective nature inherently limits conclusions, but selection bias is particularly relevant here. Only the largest tumor per patient was included, which might skew results toward worse baseline characteristics and outcomes. This decision should be justified.

There is variability in radiosurgical equipment, planning, and follow-up protocols across 14 centers, which introduces heterogeneity. This needs more explicit acknowledgment and, ideally, sensitivity analyses.

“Tumor control” is defined as regression or stability using ±20% volume change. This threshold is acceptable but should be referenced to established neuro-oncology standards or explained further.

For OS and PFS, the definition of events (especially progression) should be clarified—was progression strictly radiographic, or were clinical criteria included?

The conclusion that “SRS offers high tumor control rates with low morbidity” is reasonable, but the assertion that it is comparable to resection is speculative without a surgical control group.

Authors should avoid implying equivalence between SRS and surgery; instead, they can state that SRS appears to be a viable alternative for selected large lesions.

Adverse radiation effects are reported, but no standard grading system (e.g., CTCAE) is mentioned. Without grading, it’s difficult to compare with other series.

It is notable that a higher dose was associated with ARE, but the implications for dose selection should be more deeply discussed.

The limitations section is present but underdeveloped. Additional points to include: (A) Lack of central imaging review; (B) Potential misclassification of progression vs. pseudoprogression; (C) Absence of quality-of-life or functional outcome measures; (D) Heterogeneity in prior treatments and timing of SRS.

Minor comments:

Some sentences in the introduction are overly long and could be tightened for clarity.

Ensure consistent terminology (e.g., “HGB” vs. “hemangioblastoma”).

Abbreviations such as OS, PFS, and ARE should be defined at first mention in both the abstract and the main text.

**Do you want your identity to be public for this peer review?** For information about this choice, including consent withdrawal, please see our For information about this choice, including consent withdrawal, please see our Privacy Policy .

Reviewer #1: **Yes:** David Jaehyun ParkDavid Jaehyun Park

Reviewer #2: No

While revising your submission, please upload your figure files to the Preflight Analysis and Conversion Engine (PACE) digital diagnostic tool, https://pacev2.apexcovantage.com/ . PACE helps ensure that figures meet PLOS requirements. To use PACE, you must first register as a user. Registration is free. Then, login and navigate to the UPLOAD tab, where you will find detailed instructions on how to use the tool. If you encounter any issues or have any questions when using PACE, please email PLOS at . PACE helps ensure that figures meet PLOS requirements. To use PACE, you must first register as a user. Registration is free. Then, login and navigate to the UPLOAD tab, where you will find detailed instructions on how to use the tool. If you encounter any issues or have any questions when using PACE, please email PLOS at figures@plos.org . Please note that Supporting Information files do not need this step.. Please note that Supporting Information files do not need this step.

---

## [Author Response · Author response to Decision Letter 1]

10 Sep 2025

Safety and effectiveness of stereotactic radiosurgery for larger hemangioblastomas (>2cc): a multi-center retrospective study

Submission ID: [PONE-D-25-36099] - [EMID:4a9e5eb3f7c82381]

We take this opportunity to thank the editor and reviewers of our paper for their kind collaboration to the improvement of this manuscript. We have taken into account all the concerns raised and we have made the suggested modifications. We have implemented numerous improvements to the paper. Below we justify our replies to the suggestions made by the respected reviewers of the paper. Our responses are represented in blue.

REVIEWER 1

Reviewer #1: This manuscript presents the largest multicenter series to date evaluating stereotactic radiosurgery (SRS) for larger hemangioblastomas (>2 cc), addressing an important gap in the radiosurgical literature. The study is technically sound in its core design, with clear inclusion criteria, a well-defined patient cohort, and appropriate primary outcomes of tumor control, overall survival (OS), and progression-free survival (PFS). The statistical methods employed, including Kaplan–Meier survival analysis, log-rank testing, and Cox regression, are appropriate for the data. Results are presented in an organized manner, and the overall writing is clear and intelligible.

Response:

Thank you for your valuable feedback and the recognition of the study's clinical relevance. We appreciate the reviewer’s comments to help improve the value of our research. In this new version, we have modified our manuscript based on your invaluable suggestions.

Comment:

However, several issues limit the strength of the conclusions. The retrospective nature of the study across fourteen centers inevitably introduces heterogeneity in SRS technique, imaging follow-up, and patient selection. While the authors acknowledge variability in equipment and protocols, the absence of any adjustment for inter-center or temporal differences in dose selection, imaging frequency, or margin dose may allow confounding effects on tumor control and adverse radiation event (ARE) rates. Given that the study period spans three decades (1993–2023), practice evolution during this interval could have influenced outcomes; stratifying results by treatment era or controlling for center in multivariable analysis would strengthen the validity of the findings.

Response:

We agree and acknowledge the limitations you correctly point out. The retrospective and multicenter nature of this study inevitably introduces heterogeneity in radiosurgical technique, imaging follow-up, and patient selection across three decades. We attempted to enhance generalizability through strict inclusion criteria and a large cohort drawn from fourteen international centers. To mitigate potential confounding, we used multivariable regression analyses and demonstrated that critical radiosurgical parameters such as number of fractions, margin dose, and maximum dose were fairly balanced between baseline VHL-associated and sporadic cases (all p > 0.05). Conversely, differences in parameters such as target tumor volume and number of isocenters (p < 0.05) highlight the complexity of treating larger hemangioblastomas and underscore the importance of future studies.

Ultimately, while this study represents the largest multicenter evaluation of SRS for larger hemangioblastomas, its primary aim is to lay the groundwork for prospective, multi-institutional clinical trials that will better control for inter-center variability and strengthen the validity of our findings.

Regarding the study period, although our cohort spanned a relatively long timeframe (1993–2023), we agree that the learning curve could have influenced treatment outcomes. We attempted to perform a stratified time analysis, but this was limited by the small number of cases across eras; for example, despite the comprehensiveness of the database, only 3 of the 91 patients were treated before 2000, which precluded meaningful year-based comparisons.

We have modified the limitations section considering your thoughtful comments:

Discussion

[…]

Limitations

While this multicenter study extensively addresses a previously unexplored question using strict inclusion criteria across a large cohort, it is not without limitations inherent to its retrospective design. Despite efforts to standardize data collection, screening, and selection processes, potential selection bias, variability in data collection across centers, and differences in radiosurgical techniques remains. Although strict inclusion criteria were applied, the heterogeneity of SRS equipment and protocols, as well as the learning curve across the long study period (1993–2023), may have influenced treatment outcomes. While we achieved a robust median follow-up duration of 51.5 to 58 months, three cases were lost to follow-up. Furthermore, although the study stratifies outcomes between VHL-associated and sporadic HGB, it does not include molecular analyses, such as VHL or HIF2A mutation status, which could provide valuable insights into treatment response. . Additionally, while we implemented strict criteria for tumor assessment, isolated cyst expansion may reflect vascular permeability–driven changes rather than true progression, and the absence of stratified volumetric analysis between cystic and solid components over time may limit interpretability. Lastly, the absence of direct comparisons with surgical outcomes limits our ability to draw conclusions on the relative efficacy of SRS, particularly for surgically accessible lesions. Overall, this study provides a foundation for future prospective, multi-institutional clinical trials.”

Comment:

The definition of tumor control as either regression or stability is reasonable, but the handling of cystic enlargement is not fully clarified. Hemangioblastomas often exhibit cyst dynamics independent of solid nodule growth, and it is important to state explicitly whether isolated cyst expansion was classified as progression, particularly since this may disproportionately affect larger lesions. The omission of volumetric progression-free survival analysis in such cases may limit interpretability.

Response:

We understand your concern. In our Methods, we defined progression as either solid nodule growth, cystic enlargement, or both in combination, and this was standardized across centers to improve consistency. We recognize, however, that isolated cyst enlargement may not always represent true tumor progression, as it can result from VEGF-mediated vascular permeability, plasma ultrafiltrate leakage, or reactive peritumoral changes rather than direct tumor proliferation. Given these biological nuances, particularly in larger lesions, we have clarified our definition in the revised Methods and added to the Limitations that the absence of volumetric progression-free survival stratified by cystic versus solid components may restrict interpretability.

“Methods:

Clinical and radiological follow up:

Patients were typically followed clinically and radiographically with gadolinium-enhanced MRI every 6 months after the initial SRS and then at 6-to-12-month intervals, with further follow-up adjusted longitudinally based on individual patient needs and the protocols of each participating center. Tumor response was categorized as regression (≥20% decrease in tumor volume from baseline), stability (±20% change in tumor volume from baseline), or progression (≥20% increase in tumor, cyst volume, or both from baseline)9. Tumor control was defined as either regression or stability.

Discussion

[…]

Limitations

[…]

Additionally, while we implemented strict criteria for tumor assessment, isolated cyst expansion may reflect vascular permeability–driven changes rather than true progression, and the absence of stratified volumetric analysis between cystic and solid components over time may limit interpretability. Lastly, the absence of direct comparisons with surgical outcomes limits our ability to draw conclusions on the relative efficacy of SRS, particularly for surgically accessible lesions. Overall, this study provides a foundation for future prospective, multi-institutional clinical trials.”

Comment:

Additionally, while adverse radiation events were correlated with higher dose parameters, the finding that tumor control rates in this large-lesion cohort (~70%) are lower than historical reports for smaller lesions warrants more explicit discussion in both the abstract and conclusion, along with tempering the assertion that SRS is broadly “safe and effective” for all large lesions.

Response:

Thank you for your comment. Our use of the term “safe and effective” was intended in the context of large hemangioblastomas rather than as a direct comparison to smaller counterparts, which were beyond the scope of this study. This is particularly relevant given that a substantial proportion of large lesions in our cohort (≈20% located in the brainstem) would not be ideal candidates for surgery. Regarding adverse radiation effects (ARE), although higher dose parameters correlated with their occurrence, six of eight cases were asymptomatic, and the two symptomatic cases experienced only mild, manageable symptoms (imbalance or headache, nausea, and dizziness). Taken together, achieving ~70% tumor control with an 8% ARE rate (predominantly asymptomatic) supports the relative safety of SRS even in larger lesions. We did not intend to imply that outcomes were equivalent to smaller lesions, since these were not included in our study. We agree, however, that tumor control rates in our large-lesion cohort (~70%) are lower than historical reports for smaller hemangioblastomas, and we have revised the tone of the abstract and manuscript to reflect this distinction, tempering the assertion of SRS as broadly “safe and effective” and instead framing it within the clinical spectrum of large (>2cc) hemangioblastomas.

“Abstract:

[…]

Conclusions:

SRS may provide favorable tumor control with low morbidity in both VHL-associated and sporadic cases of larger HGB (>2 cc). Future studies should compare SRS with resection for larger HGB and explore molecular predictors of favorable response to SRS.”

Manuscript

[…]

Conclusion:

In this multi-institutional study, SRS may offer favorable tumor control rates with low morbidity in both VHL-associated and sporadic cases of larger HGB(>2cc). Comparable OS and PFS outcomes between groups highlight its viability as a minimally invasive alternative, particularly for surgically challenging lesions. Future research should explore direct comparisons with resection and incorporate molecular analyses to refine treatment strategies further.”

Comment:

The absence of a surgical comparison arm limits the ability to contextualize SRS results relative to the current gold standard for accessible lesions. While the discussion refers to SRS as a potential alternative to resection, the lack of comparative data makes this claim premature. Statements implying equivalence should be revised to emphasize that SRS appears to be a viable option for surgically challenging cases, with low morbidity and acceptable control rates, but further direct comparisons are required.

Response:

We agree with this statement and acknowledge that we did not frame this point appropriately in the original discussion. We have revised the discussion per your suggestion to clarify that our study lacked head-to-head comparisons with surgical cohorts. The revised text emphasizes that SRS appears to be a viable option, and may represent an alternative for surgically challenging cases with low morbidity and acceptable control rates, while recognizing that direct comparative studies are required before any conclusions can be made regarding equivalence with resection.

“Discussion:

[…]

In this context, our study suggest that SRS could be utilized safely, and could potentially represent a potential treatment option to resection, especially given the risks of open resection for large vascular lesions; however, this study did not make parallel assessments with surgical cohorts26. Nevertheless, direct comparisons between cerebellar and brainstem lesions treated with SRS versus resection27–29, including adjunctive measures such as embolization30,31, are essential to refine management strategies. Although SRS complications are likely limited, future studies should evaluate these outcomes to guide treatment decisions more comprehensively.”

Comment:

Similarly, although OS and PFS did not differ significantly between VHL-associated and sporadic cases, the interpretation should note that the tumor control rate in both groups still leaves a substantial proportion of patients requiring salvage treatment.

Response:

We agree with this statement. For this reason, we identified subsequent treatments required in our cohort, as summarized in Table 1 including repeat SRS, surgical resection, and fractionated radiotherapy. We have revised the Discussion to highlight that despite the potential use of SRS, the tumor control rates still left a substantial proportion of patients requiring additional treatment, underscoring the importance of considering multimodal strategies for some large lesions.

“Discussion:

[…] While supratentorial HGB have been reported to be safely treated with resection, incomplete resection is more commonly observed in some locations such as the sellar/suprasellar region38. In surgically inaccessible or eloquent areas where a gross total resection is deemed unlikely, SRS could be considered a viable treatment option. However, future studies are needed to further evaluate its efficacy and safety in managing large HGB. This is particularly significant for VHL patients, who often face repeated interventions due to multiple lesions4,39. Nevertheless, while our findings suggest that SRS may be implemented for larger HGB, the tumor control rate in both groups still leaves a substantial proportion of patients requiring salvage treatment, highlighting the potential role of multimodal approach.”

Comment:

The manuscript is generally well written, but the abstract slightly overstates generalizability and should explicitly mention its retrospective design and lack of surgical comparator. Some redundancy between the Results and Discussion could be reduced, and minor grammatical edits would further improve flow. Figures and tables are appropriate, but table formatting could be standardized for decimal precision and unit presentation.

Finally, the data availability statement indicates that access requires institutional review board approval and a request to the corresponding author. While this is understandable given patient confidentiality, it is less than optimal in meeting PLOS ONE’s open data policy, and the feasibility of depositing anonymized data in a controlled-access repository should be explored.

In summary, this study provides valuable multicenter evidence supporting the role of SRS in managing larger hemangioblastomas, especially when surgery is not feasible. The conclusions are generally supported by the data but require a more cautious interpretation in light of the retrospective design, inter-center variability, and modest tumor control rates compared with smaller lesion series. With revisions to address confounding factors, clarify cystic lesion outcome classification, and adjust the framing of conclusions, this work would make a meaningful contribution to the literature.

Response to Reviewer:

Thank you for these helpful comments. We have revised the Abstract and the Discussion to explicitly note the retrospective design and the absence of a surgical comparator. We also reduced redundancy between the Results and Discussion to improve clarity. The Limitations section was expanded to address issues of generalizability, and the Data Availability statement was revised to clarify that, due to patient confidentiality and institutional regulations across centers, raw data cannot be publicly deposited, but de-identified data may be made available upon reasonable request to the corresponding author and subject to institutional review board approval.

We sincerely thank the reviewer for these valuable contributions, w

---

## [Decision Letter · Decision Letter 1]

2 Nov 2025

Safety and effectiveness of stereotactic radiosurgery for larger hemangioblastomas (>2cc): a multi-center retrospective study

PONE-D-25-36099R1

Dear Dr. Sheehan,

We’re pleased to inform you that your manuscript has been judged scientifically suitable for publication and will be formally accepted for publication once it meets all outstanding technical requirements.

Kind regards,

Muhammad Mohsin Khan

Academic Editor

PLOS ONE

Additional Editor Comments (optional):

Dear Dr sheehan

Thank you for submitting the revised version of your manuscript to PLOS ONE. We have now received comprehensive and independent reviews following your first revision. Based on the reviewers feedback and the improvements made, I am pleased to inform you that your manuscript has been accepted for publication.

Congratulations on this achievement, and thank you for choosing PLOS ONE as the platform for your work. The production team will be in touch shortly regarding the next steps in the publication process.

With best regards,

Dr. Muhammad Mohsin Khan

Reviewers' comments:

Reviewer's Responses to Questions

**Comments to the Author**

Reviewer #2: All comments have been addressed

2. Is the manuscript technically sound, and do the data support the conclusions?

Reviewer #2: Yes

3. Has the statistical analysis been performed appropriately and rigorously?

Reviewer #2: Yes

4. Have the authors made all data underlying the findings in their manuscript fully available?

Reviewer #2: Yes

5. Is the manuscript presented in an intelligible fashion and written in standard English?

Reviewer #2: Yes

Reviewer #2: All my previous comments have been adequately addressed by the authors. I have no further concerns, and the manuscript is now suitable for publication in its current form.

**Do you want your identity to be public for this peer review?** For information about this choice, including consent withdrawal, please see our For information about this choice, including consent withdrawal, please see our Privacy Policy .

Reviewer #2: No
